# Clinical and psychosocial factors associated with domestic violence among men and women in Kandy, Sri Lanka

**Piumee Bandara**[1,2]*, **Andrew Page**[1], **Lalith Senarathna**[2,3], **Kumudu Wijewardene**[4], **Tharuka Silva**[2,5], **David Gunnell**[6,7], **Duleeka Knipe**[2,6☯], **Thilini Rajapakse**[2,5☯]

**1** Translational Health Research Institute, Western Sydney University, New South Wales, Australia, **2** South Asian Clinical Toxicology Research Collaboration, Faculty of Medicine, University of Peradeniya, Peradeniya, Sri Lanka, **3** Department of Health Promotion, Faculty of Applied Sciences, Rajarata University of Sri Lanka, Mihintale, Sri Lanka, **4** Department of Community Medicine, Faculty of Medical Sciences, University of Sri Jayawardenepura, Nugegoda, Sri Lanka, **5** Department of Psychiatry, Faculty of Medicine, University of Peradeniya, Peradeniya, Sri Lanka, **6** Population Health Sciences, Bristol Medical School, University of Bristol, Bristol, United Kingdom, **7** National Institute of Health Research Biomedical Research Centre, University Hospitals Bristol and Weston NHS Foundation Trust, Bristol, United Kingdom

☯ These authors contributed equally to this work.
* p.bandara@westernsydney.edu.au

**Data Availability Statement:** Data are available at the University of Bristol data repository, data.bris: Citation: Dee Knipe, Judi Kidger (2020): Data from

## Abstract

Domestic violence (DV) is a violation of human rights with adverse intergenerational consequences on physical and mental health. Clinical and psychosocial correlates of DV have been documented internationally, but evidence from South Asia is limited, especially among men. This is a nested cross-sectional study of the control population (N = 856) of a large case-control study in Kandy, Sri Lanka. Multivariable logistic regression models were conducted to estimate the association between clinical and psychosocial factors and experience of DV. Overall associations were examined and stratified by sex and type of abuse. Overall, 19% (95% CI 16%-21%) of the sample reported DV of any form in past year, with a similar prevalence being reported in both men (18% 95% CI 14%–22%) and women (19% 95% CI 15%–23%). Depression symptoms (adjusted OR [AOR] 3.28 95% CI 2.13–5.05), suicidal ideation (AOR 6.19 95% CI 3.67–10.45), prior diagnosis of a mental illness (AOR 3.62 95% CI 1.61–8.14), and previous self-harm (AOR 6.99 95% CI 3.65–13.38) were strongly associated with DV, as were indicators of perceived poor social support (AOR range 2.48–14.18). The presence of in-laws (AOR 2.16 95% CI 1.34–3.48), having three or more children (AOR 2.15 95% CI 1.05–4.41) and being divorced/separated/widowed were also strongly associated with DV (AOR 2.89 95% CI 1.14–7.36). There was no statistical evidence that any associations differed by sex. A multi-sectoral approach is needed to address DV in this context. Enhanced coordination between DV support services and mental health services may be beneficial. Further research and support for men as well as women is needed.

ACE & Self harm Sri Lanka (10- 2020). https://doi.
org/10.5523/bris.37pg6mv6x35r12b98aoq4blcgs.
Researchers can request access through the data
request form on the repository site. Given the high
sensitivity of the data, only researchers at verifiable
institutions will be able to access data. Any
requests will be reviewed by the University of
Bristol Access Committee, which includes senior
researchers and representatives from the
University. Data will only be released once a
controlled data access agreement has been signed
by a nominated institutional signatory.
Correspondence to Dr. Duleeka W Knipe; dee.
knipe@bristol.ac.uk.

**Funding:** This work was supported by the UK
Medical Research Council (Grant no. MC_PC_MR/
R019622/1, https://mrc.ukri.org/funding/), the
Elizabeth Blackwell Institute for Health Research,
University of Bristol (URL http://www.bristol.ac.uk/
blackwell/funding/), and the Wellcome Trust
Institutional Strategic Support Fund, through
grants awarded to DK. The funders had no role in
study design, data collection and analysis, decision
to publish, or preparation of the manuscript.

**Competing interests:** The authors have declared
that no competing interests exist.

## Introduction

Domestic violence (DV) is a serious public health issue, with longstanding intergenerational consequences. DV broadly encompasses physical, sexual, emotional and psychological abuse perpetrated by any household member. There is no current estimate of the global burden of domestic violence, however, the WHO reports that approximately a third of women have experienced violence from an intimate partner in their lifetime [1]. Rates are particularly high in South Asia, where 42% of women reported past-year intimate partner violence (IPV), compared to 23% in high-income countries [1]. This estimate does not take into consideration abuse experienced by men or non-partner abuse, despite high rates of parental and in-law violence, reported in this context [2, 3]. Within Sri Lanka, it is estimated 40% (95% CI 38% - 42%) of women aged 15 years or older have experienced physical, sexual, emotional, and/or economic violence and/or controlling behaviours by a partner in their lifetime [4].

To effectively inform DV prevention and appropriate management and support it is crucial to identify its associated factors. There is strong evidence of a bi-directional relationship between mood disorders and IPV and DV globally [5–8]. Studies have also reported complex interactions between DV, mood disorders, and psychosocial wellbeing, with evidence that social support may alleviate the adverse mental health impacts of abuse [9–11]. Despite strong international evidence of links with DV, the psychosocial and mental health profile of men and women who experience DV is poorly understood in South Asia. In addition, there is a scarcity of evidence on the correlates of DV among men globally. Notably, a WHO multi-country study showed that within Sri Lanka, men and women reported a similar prevalence of past-year IPV, signalling the importance of further research among men as well as women [12]. Furthermore, researchers have highlighted the need to explore different types of abuse [12]. DV research from South Asia has largely concentrated on physical abuse [13] and thus little is known in this context about the level and correlates of more covert forms of abuse such as psychological abuse.

Accordingly, the aims of this study were to examine 1) the prevalence of any DV and different types of DV (physical/sexual abuse and psychological abuse) overall and for men and women from the source population (Kandy); 2) clinical (e.g. depression symptoms, alcohol misuse, and suicidal behaviour) and psychosocial factors (e.g. social support and household composition) associated with DV in Kandy, Sri Lanka; and 3) to explore how these associations may differ by sex and by type of abuse.

## Methods

### Study setting

The control series from a large case-control study examining childhood adversity and deliberate self-poisoning in Kandy, Sri Lanka, was used for this study [14, 15]. The Kandy District is situated in the Central highland province of the island, approximately 115 kilometres from the nation's capital, Colombo. Kandy is characterised as a key cultural, administrative and commercial centre. It is densely populated with approximately 1.4 million people, of which 81% live in rural areas, 12% urban and 6% in the plantation sector [16]. The majority of people identify as Sinhalese (74%), followed by Moor (14%), and Tamil (11%). Buddhism is the dominant religion in Kandy (74%) and throughout Sri Lanka (70%) [16].

### Data collection

Adults (≥18 years) frequency matched on age and sex to self-poisoning cases were recruited from two sources: (i) the outpatient department of a tertiary hospital (Teaching Hospital

Peradeniya), and (ii) from households within the hospital catchment area within Kandy district. Hospital-based participants were defined as outpatients (27%) or accompanying visitors (73%), hereafter referred to as 'bystanders', presenting to the outpatient department between July 2018 to December 2018. To address the possibility that hospital controls may not be representative of the population giving rise to the cases we additionally recruited door-to-door from twelve randomly selected villages (*Grama Niladhari* sub-divisions) within the main population catchment of the hospital from January to April 2019. Selected villages were compared with 2017 Census data to ensure similar distributions to the source population in terms of sex, age, and ethnicity. Due to resource constraints and logistical challenges (e.g. topography of the region), not every household could be reached within the sampling frame. For every household approached, only one participant was selected for interview. If more than one participant was eligible, the participant with the most recent birthday was interviewed.

All interviews were conducted by trained data collectors with a nursing or basic science degree in the participant's preferred language (Sinhala, Tamil or English). The research was explained to each participant and formal written consent was obtained. All interviews were undertaken in private to ensure responses would not be influenced by another person and for patient safety. Participants in any location who could not be interviewed in private, or who were physically or cognitively unable to participate, were not eligible for interview. Participants who had previously self-harmed in their lifetime, although excluded in the analysis of the broader case-control study [15], were included in the current study, therefore numbers will differ from other publications using the dataset.

A detailed description of ethical considerations and safeguards is outlined in the study protocol for the case-control study [14]. In brief, participants who disclosed daily suicidal thoughts within the preceding two weeks were referred to the psychiatry clinic (Teaching Hospital Peradeniya) for further management and follow up. Participants who disclosed DV were offered discrete contact information for counselling support and referred to the psychiatric clinic, if appropriate. All research was approved by the Ethical Review Committee of the Faculty of Medicine, University of Peradeniya, Sri Lanka (14 June 2018) and conformed to the principles embodied in the Declaration of Helsinki.

Prior analyses comparing hospital-based and household-based controls showed similarities in terms of sociodemographic and clinical characteristics [17]. Given hospital bystanders did not present for clinical concerns, and notionally reflect members of the community, a decision was made to exclude outpatients only i.e. individuals presenting to hospital for health care (n = 144) *a priori* [18], and combine the bystander and household controls into one sample to enhance statistical power.

## Study variables

**Outcome variables.** Data on the outcome DV was collected using the Humiliation, Afraid, Rape, Kick (HARK) questionnaire which has been previously shown to identify partner violence with high specificity (95%) and sensitivity (81%) [19]. The HARK tool identifies four types of abuse–physical, sexual, humiliation/emotional abuse, and fear of an intimate partner in the past year. The questionnaire was broadened to include past-year abuse by any family member living in the household, not just by an intimate partner (S1 Table). The tool was then translated, back-translated and piloted in the two local languages (Sinhala and Tamil) with individuals in the outpatient department. No modifications were required after piloting with the local population. A HARK score of ≥1 indicates an experience of at least one form of past-year DV. To distinguish effects by type of violence, outcome variables were defined for physical/sexual abuse and psychological abuse. Psychological abuse was categorised as individuals

who reported experiencing fear of any family member living in the household and/or humiliation (without any physical or sexual violence) in the past year, versus no abuse. A composite physical/sexual abuse variable (with or without psychological abuse) was created given the limited number of total sexual violence cases (n = 4).

**Other study factors.** All study questionnaires and instruments used in this study (in English and local languages) can be accessed upon request from the University of Bristol data depository [20]. Sociodemographic data on age, ethnicity, education level, marital status, number of children, and household composition, were collected using a questionnaire pretested with outpatients and bystanders from the outpatient department. Depression symptoms in the last two weeks were measured using the nine-item Patient Health Questionnaire (PHQ-9) with a cut-off score of ≥10. The PHQ-9 is validated for use within the Sri Lankan population [21]. The ninth item of the PHQ-9 examining any suicidal ideation over the last two weeks, was included in the study as a separate variable in a post-hoc analysis. Prior diagnosis of a mental disorder from a health professional, lifetime previous self-harm, and presence of chronic illness and/or disability were ascertained through participant self-report. Harmful alcohol use was measured using the Alcohol Use Disorders Identification Test (AUDIT), with a cut-off score of ≥8 denoting hazardous drinking. The AUDIT has been validated for use within the Sri Lankan population [22]. Questions relating to social support were derived from a large social capital community survey in the North Central Province of Sri Lanka [23]. Participants were asked to rate on a five-item Likert scale to what extent they agreed or disagreed with statements relating to social and emotional support at the household and community level. Each of the four items were categorised into a binary (agree vs. disagree) variable, with the small number of neutral responses combined with 'strongly agree/tend to agree' responses (S2 Table).

## Statistical analysis

The statistical analysis plan was specified *a priori* [18]. Although the original analysis plan stated that all analyses would be presented stratified by sex only, due to low numbers and low statistical precision, a decision was subsequently made to present associations both using the overall sample, and stratified by sex. In addition, suicidal ideation was also examined in a post-hoc analysis. All analyses were conducted on complete data. Differences in the frequency and distribution of all study factors between bystanders and household-based participants were analysed and tested for heterogeneity using chi squared tests. The association between clinical factors (depression symptoms, suicidal ideation, alcohol misuse, prior psychiatric disorder diagnosis, previous self-harm, chronic illness/disability) and psychosocial factors (perceived social support, household composition–civil status, number of children, family structure) and any DV were assessed overall using a series of adjusted logistic regression models, then stratified by sex. To examine effect modification by sex for other co-variates, likelihood ratio tests were conducted to compare model fit for models with, or without, an interaction term between sex and a given co-variate. Given that age has been previously shown to be associated with DV [13, 24] and psychiatric morbidity [25], and is hypothesised to be linked to psychosocial factors in this context, all models were adjusted for the confounder age. Ethnic minority groups in Sri Lanka are more likely to experience political violence, marginalisation and social disadvantage [26], which is likely to impact their level of community social support and DV. In addition, a complex range of socio-cultural factors across ethnic groups may influence household composition (e.g. number of children and family structure). Therefore, models for the association between psychosocial factors and DV additionally adjusted for ethnicity.

A second model (Model 2) was then fitted additionally adjusting for education level, for all study exposures examined. Lower educational attainment is known to be associated with household size, clinical and psychosocial factors, and DV in LMIC settings [27–30]. However, given lower educational attainment may also be a consequence of factors, such as a diagnosis of a psychiatric disorder, education level was not included in the main model (Model 1). The differential associations by type of abuse (physical/sexual abuse and psychological abuse) were explored in a supplementary analysis using logistic regression models (as classified above). Due to limitations in case numbers, this analysis was not stratified by sex.

Sensitivity analyses were conducted using a sample restricted to household-based community controls to measure the extent to which associations differ between the combined household and bystander sample and household only sample. All regression analyses were conducted using the 'logistic' command in Stata (version 15.1, Stata Corp, College Station, TX, USA).

## Results

A total of 382 hospital bystanders and 480 household-based participants were interviewed (N = 862). Sixteen participants (<2%; 11 bystanders, 5 household) were excluded due to missing data, resulting in a combined sample of 846 adults. The response rate for hospital-based and household-based participants was equivalent (64%). Overall, females were more likely to respond than males. No further data were collected on non-respondents. No statistical differences were found between household-based participants and hospital bystanders on DV prevalence, sex, age, education level, clinical, and social support factors, or on the presence of in-laws or extended family (Table 1). Compared to hospital bystanders, household-based participants had a higher representation of ethnic minorities (p = 0.03), and were more likely to be married (p = 0.01), have children (p = 0.003), and live in a nuclear household (p = 0.03) (Table 1).

Characteristics of the combined sample (N = 846) are presented in Table 1. The overall prevalence of DV of any form in the past year was 19% (95% CI 16%–21%), and higher for psychological abuse without any other form of abuse (16% 95% CI 13%– 18%) compared to physical/sexual abuse, with or without any other form of abuse (4% 95% CI 3%–6%) (Table 1). Similar rates of any DV were found for women (19% 95% CI 15%–23%) and men (18% 95% CI 14%–22%). Examination of different forms of abuse showed similar prevalence for women and men for psychological abuse (women: 16% 95% CI 13% - 19%; men: 15% 95% CI 12% - 20%) and physical/sexual abuse (women: 5% 95% CI 3% - 7%; men: 3% 95% CI 2% - 6%). Sociodemographic factors identified *a priori* as potential confounders (age, ethnicity, educational attainment) were evenly distributed between DV and non-DV cases (S3 Table).

After adjusting for confounders, consistent associations between depression symptoms (OR 3.28 95% CI 2.13–5.05), suicidal ideation (OR 6.19 95% CI 3.67–10.45), prior diagnosis of a mental illness (OR 3.62 95% CI 1.61–8.14), previous self-harm (OR 6.99 95% CI 3.65–13.38), chronic illness/disability (OR 1.65 95% CI 1.04–2.62) and experience of DV were found but not for harmful alcohol use (Table 2, Fig 1). Indicators of perceived poor social support i.e. not feeling supported in difficult situations (OR 14.18 95% CI 6.02–33.40), not being able to share joy and grief with a household member (OR 9.97 95% CI 4.82–20.61) or community member (OR 2.48 95% CI 1.59–3.87), and not feeling at home in community (OR 3.08 95% CI 1.86–5.10) were strongly associated with DV. Finally, presence of in-laws (OR 2.16 95% CI 1.34–3.48), having three or more children (OR 2.15 95% CI 1.05–4.41) and being divorced, separated or widowed were also associated with DV (OR 2.89 95% CI 1.14–7.36) (Table 2, Fig 1).

After stratification by sex, point estimates for clinical and psychosocial factors (especially household social support), were larger for women compared to men. The exception to this was

**Table 1. Distribution of study characteristics by recruitment source and overall.**

| | Recruitment source | | | | | Total (N = 846) | |
| | Hospital bystander (n = 371) | | Household (n = 475) | | | | |
| | N | % | N | % | P value[a] | N | % |
|---|---|---|---|---|---|---|---|
| **Domestic violence** | | | | | | | |
| Any domestic violence | 73 | 19.7 | 84 | 17.7 | 0.46 | 157 | 18.6 |
| Physical/sexual violence | 12 | 3.9 | 17 | 4.2 | 0.84 | 29 | 4.0 |
| Psychological violence only | 61 | 17.0 | 67 | 14.6 | 0.36 | 128 | 15.7 |
| **Sex** | | | | | | | |
| Male | 151 | 40.7 | 207 | 43.6 | 0.40 | 358 | 42.3 |
| Female | 220 | 59.3 | 268 | 56.4 | | 488 | 57.7 |
| **Age** | | | | | | | |
| 18 to 30 | 233 | 62.8 | 269 | 56.6 | 0.11 | 502 | 59.3 |
| 31 to 45 | 85 | 22.9 | 115 | 24.2 | | 200 | 23.6 |
| 46 to 90 | 53 | 14.3 | 91 | 19.2 | | 144 | 17.0 |
| **Ethnicity** | | | | | | | |
| Sinhala | 336 | 90.6 | 406 | 85.5 | 0.03 | 742 | 87.7 |
| Non-Sinhala | 35 | 9.4 | 69 | 14.5 | | 104 | 12.3 |
| **Highest education level** | | | | | | | |
| Passed A/L or completed tertiary | 192 | 51.8 | 226 | 47.6 | 0.43 | 418 | 49.4 |
| Passed O/L | 92 | 24.8 | 134 | 28.2 | | 226 | 26.7 |
| Completed between grades 1–10, or no schooling | 87 | 23.5 | 115 | 24.2 | | 202 | 23.9 |
| **Depression symptoms (PHQ-9≥10)** | | | | | | | |
| No | 316 | 85.2 | 417 | 87.8 | 0.27 | 733 | 86.6 |
| Yes | 55 | 14.8 | 58 | 12.2 | | 113 | 13.4 |
| **Any suicidal ideation (PHQ item 9)** | | | | | | | |
| No | 345 | 93.0 | 435 | 91.6 | 0.45 | 780 | 92.2 |
| Yes | 26 | 7.0 | 40 | 8.4 | | 66 | 7.8 |
| **Ever diagnosed with mental illness** | | | | | | | |
| No | 364 | 98.1 | 457 | 96.2 | 0.11 | 821 | 97.0 |
| Yes | 7 | 1.9 | 18 | 3.8 | | 25 | 3.0 |
| **Previously self-harmed** | | | | | | | |
| No | 355 | 95.7 | 450 | 94.7 | 0.52 | 805 | 95.2 |
| Yes | 16 | 4.3 | 25 | 5.3 | | 41 | 4.8 |
| **Harmful alcohol use (AUDIT≥8)** | | | | | | | |
| No | 331 | 89.2 | 413 | 86.9 | 0.31 | 744 | 87.9 |
| Yes | 40 | 10.8 | 62 | 13.1 | | 102 | 12.1 |
| **Chronic illness/disability** | | | | | | | |
| No | 324 | 87.3 | 403 | 84.8 | 0.30 | 727 | 85.9 |
| Yes | 47 | 12.7 | 72 | 15.2 | | 119 | 14.1 |
| **Household member to share joy and grief** | | | | | | | |
| Yes | 357 | 96.2 | 454 | 95.6 | 0.64 | 811 | 95.9 |
| No | 14 | 3.8 | 21 | 4.4 | | 35 | 4.1 |
| **Household member supportive in difficult situations** | | | | | | | |
| Yes | 363 | 97.8 | 455 | 95.8 | 0.10 | 818 | 96.7 |
| No | 8 | 2.2 | 20 | 4.2 | | 28 | 3.3 |
| **Community member to share joy and grief** | | | | | | | |
| Yes | 317 | 85.4 | 417 | 87.8 | 0.32 | 734 | 86.8 |

*(Continued)*

**Table 1.** (Continued)

| | Recruitment source | | | | | Total (N = 846) | |
|---|---|---|---|---|---|---|---|
| | Hospital bystander (n = 371) | | Household (n = 475) | | | | |
| | N | % | N | % | P value[a] | N | % |
| No | 54 | 14.6 | 58 | 12.2 | | 112 | 13.2 |
| **Feel at home in community** | | | | | | | |
| Yes | 342 | 92.2 | 426 | 89.7 | 0.21 | 768 | 90.8 |
| No | 29 | 7.8 | 49 | 10.3 | | 78 | 9.2 |
| **Civil status** | | | | | | | |
| Married | 177 | 47.7 | 265 | 55.8 | 0.01 | 442 | 52.2 |
| Unmarried | 187 | 50.4 | 194 | 40.8 | | 381 | 45.0 |
| Divorced | 7 | 1.9 | 16 | 3.4 | | 23 | 2.7 |
| **Number of children** | | | | | | | |
| None | 220 | 59.3 | 231 | 48.6 | 0.003 | 451 | 53.3 |
| One to two | 105 | 28.3 | 186 | 39.2 | | 291 | 34.4 |
| Three or more | 46 | 12.4 | 58 | 12.2 | | 104 | 12.3 |
| **Nuclear family** | | | | | | | |
| No | 161 | 43.4 | 241 | 50.7 | 0.03 | 402 | 47.5 |
| Yes | 210 | 56.6 | 234 | 49.3 | | 444 | 52.5 |
| **Presence of in-laws** | | | | | | | |
| No | 332 | 89.5 | 411 | 86.5 | 0.19 | 743 | 87.8 |
| Yes | 39 | 10.5 | 64 | 13.5 | | 103 | 12.2 |
| **Extended family (biological)** | | | | | | | |
| No | 352 | 94.9 | 445 | 93.7 | 0.46 | 797 | 94.2 |
| Parent/grandparent/grandchild | 19 | 5.1 | 30 | 6.3 | | 49 | 5.8 |

A/L = Advanced Level; O/L = Ordinary Level; PHQ = Patient Health Questionnaire; AUDIT = Alcohol Use Disorders Identification Test.

[a]Chi-squared test.

chronic illness/disability and poor community social support, which showed stronger associations among men (Tables 3 and 4, Fig 2). However, there was no strong statistical evidence that any associations differed by sex (Tables 3 and 4). Overall associations were largely consistent with the main model 1 after adjusting for potential confounder, educational attainment (Table 2). Minor attenuation was found among women after adjusting for education level, and to a lesser extent among men (Tables 3 and 4).

Stratification by type of abuse showed consistent associations with clinical and psychosocial factors for both physical/sexual abuse and psychological abuse (S4 Table). Given the lower case numbers, evidence was weaker for physical/sexual abuse, despite large point estimates (S4 Table). Sensitivity analyses restricted to household controls showed a similar pattern in the direction of associations compared to the main analysis. However, given the smaller sample, confidence intervals were wider and estimates were less precise but consistent with the main findings (S5 Table).

## Discussion

DV is a complex public health issue influenced by a range of factors operating at the individual, family, community and societal level. The current study sought to highlight the prevalence of DV among men and women in Kandy, Sri Lanka and the clinical, especially mental health,

**Table 2. Clinical and psychosocial factors associated with domestic violence (DV) in Kandy, Sri Lanka.**

| | DV (n = 157) | No DV (n = 689) | Model 1 | Model 2 |
|---|---|---|---|---|
| | N (%) | N (%) | OR (95% CI) | OR (95% CI) |
| *Clinical factors* | | | | |
| **Depression symptoms (PHQ-9≥10)** | | | | |
| No | 114 (72.6) | 619 (89.8) | 1.00 | 1.00 |
| Yes | 43 (27.4) | 70 (10.2) | 3.28 (2.13–5.05) | 3.14 (2.03–4.85) |
| **Any suicidal ideation (PHQ item 9)** | | | | |
| No | 122 (77.7) | 658 (95.5) | 1.00 | 1.00 |
| Yes | 35 (22.3) | 31 (4.5) | 6.19 (3.67–10.45) | 5.88 (3.47–9.97) |
| **Ever diagnosed with mental illness** | | | | |
| No | 146 (93.0) | 675 (98.0) | 1.00 | 1.00 |
| Yes | 11 (7.0) | 14 (2.0) | 3.62 (1.61–8.14) | 3.70 (1.64–8.38) |
| **Previously self-harmed** | | | | |
| No | 133 (84.7) | 672 (97.5) | 1.00 | 1.00 |
| Yes | 24 (15.3) | 17 (2.5) | 6.99 (3.65–13.38) | 6.61 (3.43–12.74) |
| **Harmful alcohol use (AUDIT≥8)** | | | | |
| No | 138 (87.9) | 606 (88.0) | 1.00 | 1.00 |
| Yes | 19 (12.1) | 83 (12.0) | 1.05 (0.61–1.79) | 0.99 (0.57–1.69) |
| **Chronic illness/disability** | | | | |
| No | 127 (80.9) | 600 (87.1) | 1.00 | 1.00 |
| Yes | 30 (19.1) | 89 (12.9) | 1.65 (1.04–2.62) | 1.66 (1.05–2.64) |
| *Social support factors* | | | | |
| **Household member to share joy and grief** | | | | |
| Yes | 134 (85.4) | 677 (98.3) | 1.00 | 1.00 |
| No | 23 (14.6) | 12 (1.7) | 9.97 (4.82–20.61) | 9.59 (4.62–19.92) |
| **Household member supportive in difficult situations** | | | | |
| Yes | 137 (87.3) | 681 (98.8) | 1.00 | 1.00 |
| No | 20 (12.7) | 8 (1.2) | 14.18 (6.02–33.40) | 13.25 (5.60–31.36) |
| **Community member to share joy and grief** | | | | |
| Yes | 120 (76.4) | 614 (89.1) | 1.00 | 1.00 |
| No | 37 (23.6) | 75 (10.9) | 2.48 (1.59–3.87) | 2.50 (1.59–3.91) |
| **Feel at home in community** | | | | |
| Yes | 128 (81.5) | 640 (92.9) | 1.00 | 1.00 |
| No | 29 (18.5) | 49 (7.1) | 2.93 (1.78–4.83) | 3.08 (1.86–5.10) |
| *Household composition* | | | | |
| **Civil status** | | | | |
| Married | 80 (51.0) | 362 (52.5) | 1.00 | 1.00 |
| Never married | 69 (43.9) | 312 (45.3) | 0.74 (0.47–1.16) | 0.82 (0.52–1.30) |
| Divorced, separated or widowed | 8 (5.1) | 15 (2.2) | 2.89 (1.14–7.36) | 2.79 (1.09–7.13) |
| **Number of children** | | | | |
| None | 82 (52.2) | 369 (53.6) | 1.00 | 1.00 |
| One to two | 53 (33.8) | 238 (34.5) | 1.48 (0.90–2.44) | 1.32 (0.79–2.20) |
| Three or more | 22 (14.0) | 82 (11.9) | 2.15 (1.05–4.41) | 1.78 (0.84–3.76) |
| **Nuclear family** | | | | |
| No | 82 (52.2) | 320 (46.4) | 1.00 | 1.00 |
| Yes | 75 (47.8) | 369 (53.6) | 0.79 (0.55–1.11) | 0.81 (0.57–1.15) |
| **Presence of in-laws** | | | | |
| No | 127 (80.9) | 616 (89.4) | 1.00 | 1.00 |

*(Continued)*

**Table 2.** (Continued)

|  | DV (n = 157) | No DV (n = 689) | Model 1 | Model 2 |
|---|---|---|---|---|
|  | N (%) | N (%) | OR (95% CI) | OR (95% CI) |
| Yes | 30 (19.1) | 73 (10.6) | 2.16 (1.34–3.48) | 2.10 (1.30–3.39) |
| **Extended family (biological)** |  |  |  |  |
| No | 150 (95.5) | 647 (93.9) | 1.00 | 1.00 |
| Parent/grandparent/grandchild | 7 (4.5) | 42 (6.1) | 0.68 (0.30–1.56) | 0.66 (0.29–1.52) |

OR = Odds ratio; CI = Confidence Interval.

Model 1: Clinical factors adjusted for age; household and social support factors adjusted for age and ethnicity.

Model 2: Additionally adjusting for educational attainment.

and psychosocial correlates of DV. A key finding was that men and women experienced a similar prevalence of DV in Kandy. Strong and consistent associations between current depression symptoms, suicidal ideation, previous self-harm, prior diagnosis of psychiatric disorder, chronic illness/disability, perceived low social support and DV were found overall. The presence of in-laws, having three or more children, and being divorced, separated or widowed were also associated with DV. There was no strong statistical evidence that associations differed by sex for almost all study variables, although there was some weak statistical evidence that lower social support in the household was associated with greater DV in women compared to men, and this was in the opposite direction for community support.

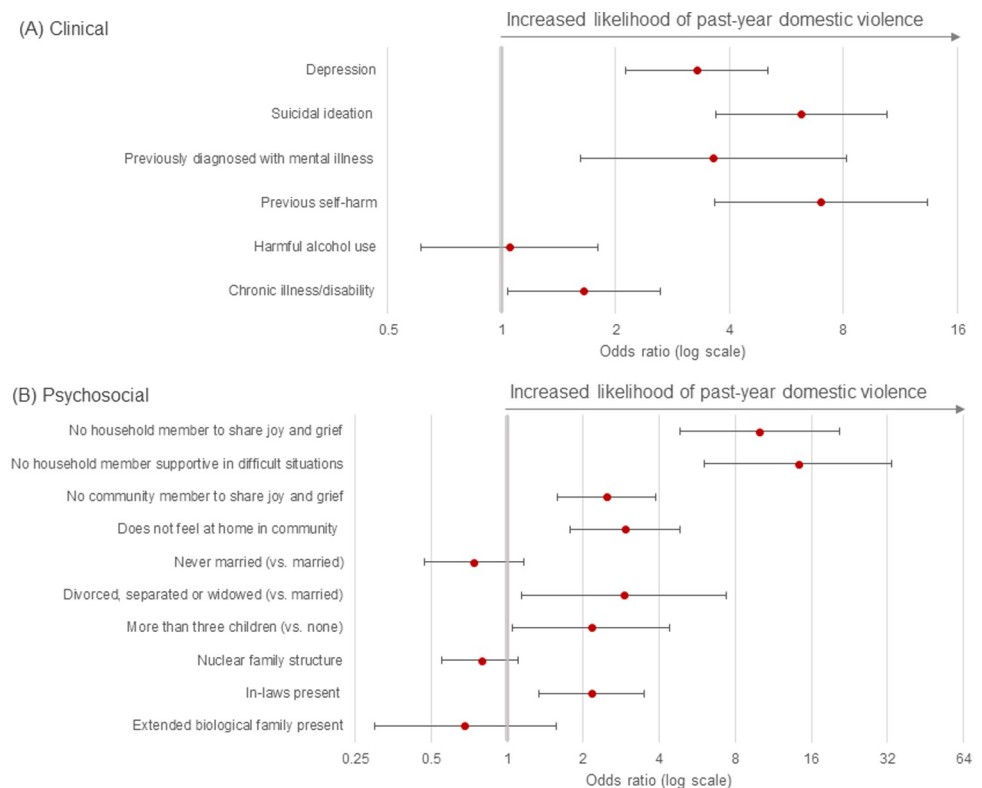

**Fig 1. Clinical and psychosocial factors associated with past-year domestic violence, Kandy, Sri Lanka.** (A) Clinical correlates of past-year domestic violence, adjusted for age. (B) Psychosocial correlates of past-year domestic violence, adjusted for age and ethnicity. The bold line indicates a null result.

**Table 3. Clinical factors associated with domestic violence (DV) in Kandy, Sri Lanka, stratified by sex.**

| | Females (n = 488) | | | | Males (n = 358) | | | | Interaction P value |
|---|---|---|---|---|---|---|---|---|---|
| | DV (n = 93) | No DV (n = 395) | Model 1 | Model 2 | DV (n = 64) | No DV (n = 294) | Model 1 | Model 2 | |
| | N (%) | N (%) | OR (95% CI) | OR (95% CI) | N (%) | N (%) | OR (95% CI) | OR (95% CI) | |
| *Clinical factors* | | | | | | | | | |
| **Depression symptoms (PHQ-9≥10)** | | | | | | | | | |
| No | 63 (67.7) | 354 (89.6) | 1.00 | 1.00 | 51 (79.7) | 265 (90.1) | 1.00 | 1.00 | |
| Yes | 30 (32.3) | 41 (10.4) | 4.03 (2.33–6.94) | 3.85 (2.22–6.67) | 13 (20.3) | 29 (9.9) | 2.31 (1.12–4.76) | 2.22 (1.07–4.64) | 0.21 |
| **Any suicidal ideation (PHQ item 9)** | | | | | | | | | |
| No | 67 (72.04) | 375 (94.9) | 1.00 | 1.00 | 55 (85.9) | 283 (96.3) | 1.00 | 1.00 | |
| Yes | 26 (28.0) | 20 (5.1) | 7.56 (3.96–14.43) | 7.19 (3.75–13.77) | 9 (14.1) | 11 (3.7) | 4.26 (1.68–10.82) | 4.04 (1.56–10.44) | 0.32 |
| **Ever diagnosed with mental illness** | | | | | | | | | |
| No | 87 (93.5) | 389 (98.5) | 1.00 | 1.00 | 59 (92.2) | 286 (97.3) | 1.00 | 1.00 | |
| Yes | 6 (6.5) | 6 (1.5) | 4.68 (1.46–15.02) | 4.48 (1.39–14.45) | 5 (7.8) | 8 (2.7) | 3.04 (0.96–9.66) | 3.27 (1.02–10.50) | 0.64 |
| **Previously self-harmed** | | | | | | | | | |
| No | 79 (84.9) | 388 (98.2) | 1.00 | 1.00 | 54 (84.4) | 284 (96.6) | 1.00 | 1.00 | 0.31 |
| Yes | 14 (15.1) | 7 (1.8) | 9.72 (3.79–24.90) | 9.16 (3.54–23.66) | 10 (15.6) | 10 (3.4) | 5.04 (1.98–12.84) | 5.05 (1.95–13.09) | |
| **Harmful alcohol use (AUDIT≥8)** | | | | | | | | | |
| No | 93 (100.0) | 393 (99.5) | 1.00 | 1.00 | 45 (70.3) | 213 (72.4) | 1.00 | 1.00 | |
| Yes | a | a | b | b | 19 (29.7) | 81 (27.6) | 1.12 (0.62–2.03) | 1.10 (0.60–2.01) | (–) |
| **Chronic illness/disability** | | | | | | | | | |
| No | 78 (83.9) | 339 (85.8) | 1.00 | 1.00 | 49 (76.6) | 261 (88.8) | 1.00 | 1.00 | |
| Yes | 15 (16.1) | 56 (14.2) | 1.20 (0.64–2.24) | 1.18 (0.63–2.22) | 15 (23.4) | 33 (11.2) | 2.52 (1.26–5.02) | 2.54 (1.27–5.08) | 0.12 |

OR = Odds ratio; CI = Confidence Interval.

[a] To avoid statistical disclosure, low counts (<5) are not shown

[b] Too few cases for calculation.

Model 1: Clinical factors adjusted for age; household and social support factors adjusted for age and ethnicity.

Model 2: Additionally adjusting for educational attainment.

The prevalence of DV in the past-year among women in the present study (19% 95% CI 16% - 21%) was similar to past-year national IPV estimates from the 2019 Women's Wellbeing Survey (15% 95% CI 13% - 16%) and 2016 DHS (17% 95% CI 16% -18%), and DHS estimate for the district of Kandy (25% 95% CI 22% - 29%). Notably, the prevalence of DV was similar for men and women. This is consistent with a previous WHO multi-country (including Sri Lanka) study on women's and men's reports of past-year IPV [12]. Studies from the UK have reported emotional abuse as the most common form of DV experienced by men in the past year, consistent with the current study [31, 32]. There is a dearth of qualitative research among men who have experienced DV in South Asia. Given the prevalence of DV among men was similar to women, further qualitative research is needed to understand how men differentially experience DV compared to women.

**Table 4. Psychosocial factors associated with domestic violence (DV) in Kandy, Sri Lanka, stratified by sex.**

| | Females (n = 488) | | | | Males (n = 358) | | | | |
|---|---|---|---|---|---|---|---|---|---|
| | DV (n = 93) | No DV (n = 395) | Model 1 | Model 2 | DV (n = 64) | No DV (n = 294) | Model 1 | Model 2 | Interaction P value |
| *Social support factors* | N (%) | N (%) | OR (95% CI) | OR (95% CI) | N (%) | N (%) | OR (95% CI) | OR (95% CI) | |
| **Household member to share joy and grief** | | | | | | | | | |
| Yes | 79 (84.9) | 392 (99.2) | 1.00 | 1.00 | 55 (85.9) | 285 (96.9) | 1.00 | 1.00 | |
| No | 14 (15.1) | 3 (0.8) | 24.09 (6.70–86.68) | 23.7 (6.55–85.79) | 9 (14.1) | 9 (3.1) | 5.39 (2.03–14.33) | 5.15 (1.92–13.83) | 0.05 |
| **Household member supportive in difficult situations** | | | | | | | | | |
| Yes | 80 (86.0) | 393 (99.5) | 1.00 | 1.00 | 57 (89.1) | 288 (98.0) | 1.00 | 1.00 | |
| No | 13 (14.0) | * | 37.03 (8.02–171.01) | 33.12 (7.13–153.76) | 7 (10.9) | 6 (2.0) | 6.48 (2.05–20.53) | 6.30 (1.98–20.05) | 0.06 |
| **Community member to share joy and grief** | | | | | | | | | |
| Yes | 74 (79.6) | 346 (87.6) | 1.00 | 1.00 | 46 (71.9) | 268 (91.2) | 1.00 | 1.00 | |
| No | 19 (20.4) | 49 (12.4) | 1.77 (0.97–3.20) | 1.85 (1.01–3.38) | 18 (28.1) | 26 (8.8) | 3.94 (1.98–7.81) | 3.88 (1.95–7.72) | 0.07 |
| **Feel at home in community** | | | | | | | | | |
| Yes | 81 (87.1) | 369 (93.4) | 1.00 | 1.00 | 47 (73.4) | 271 (92.2) | 1.00 | 1.00 | |
| No | 12 (12.9) | 26 (6.6) | 2.11 (1.02–4.37) | 2.20 (1.05–4.61) | 17 (26.6) | 23 (7.8) | 4.13 (2.04–8.34) | 4.22 (2.07–8.59) | 0.17 |
| *Household composition* | | | | | | | | | |
| **Civil status** | | | | | | | | | |
| Married | 48 (51.6) | 204 (51.6) | 1.00 | 1.00 | 32 (50.0) | 158 (53.7) | 1.00 | 1.00 | |
| Never married | 40 (43.0) | 182 (46.1) | 0.67 (0.39–1.16) | 0.77 (0.44–1.37) | 29 (45.3) | 130 (44.2) | 0.92 (0.40–2.10) | 1.03 (0.44–2.37) | |
| Divorced, separated or widowed | 5 (5.4) | 9 (2.3) | 2.94 (0.83–10.45) | 2.79 (0.78–10.0) | * | 6 (2.0) | 2.77 (0.64–12.05) | 2.81 (0.64–12.26) | 0.96 |
| **Number of children** | | | | | | | | | |
| None | 48 (51.6) | 220 (55.7) | 1.00 | 1.00 | 34 (53.1) | 149 (50.7) | 1.00 | 1.00 | |
| One to two | 33 (35.5) | 134 (33.9) | 1.75 (0.97–3.15) | 1.48 (0.79–2.78) | 20 (31.3) | 104 (35.4) | 1.03 (0.41–2.6) | 0.93 (0.37–2.36) | |
| Three or more | 12 (12.9) | 41 (10.4) | 2.74 (1.07–6.99) | 2.07 (0.76–5.65) | 10 (15.6) | 41 (13.9) | 1.49 (0.47–4.78) | 1.32 (0.41–4.29) | 0.93 |
| **Nuclear family** | | | | | | | | | |
| No | 52 (55.9) | 178 (45.1) | 1.00 | 1.00 | 30 (46.9) | 142 (48.3) | 1.00 | 1.00 | |
| Yes | 41 (44.1) | 217 (54.9) | 0.66 (0.42–1.04) | 0.68 (0.43–1.08) | 34 (53.1) | 152 (51.7) | 1.04 (0.60–1.80) | 1.07 (0.62–1.86) | 0.20 |
| **Presence of in-laws** | | | | | | | | | |
| No | 72 (77.4) | 347 (87.8) | 1.00 | 1.00 | 55 (85.9) | 269 (91.5) | 1.00 | 1.00 | |
| Yes | 21 (22.6) | 48 (12.2) | 2.33 (1.29–4.19) | 2.10 (1.15–3.82) | 9 (14.1) | 25 (8.5) | 1.94 (0.83–4.58) | 2.08 (0.87–4.97) | 0.85 |
| **Extended family (biological)** | | | | | | | | | |
| No | 88 (94.6) | 365 (92.4) | 1.00 | 1.00 | 62 (96.9) | 282 (95.9) | 1.00 | 1.00 | |
| Parent/grandparent/grandchild | 5 (5.4) | 30 (7.6) | 0.64 (0.24–1.72) | 0.61 (0.23–1.63) | * | 12 (4.1) | 0.78 (0.17–3.64) | 0.82 (0.18–3.84) | 0.91 |

OR = Odds ratio; CI = Confidence Interval.

*To avoid statistical disclosure, low counts (<5) are not shown.

Model 1: Clinical factors adjusted for age; household and social support factors adjusted for age and ethnicity. Model 2: Additionally adjusting for educational attainment.

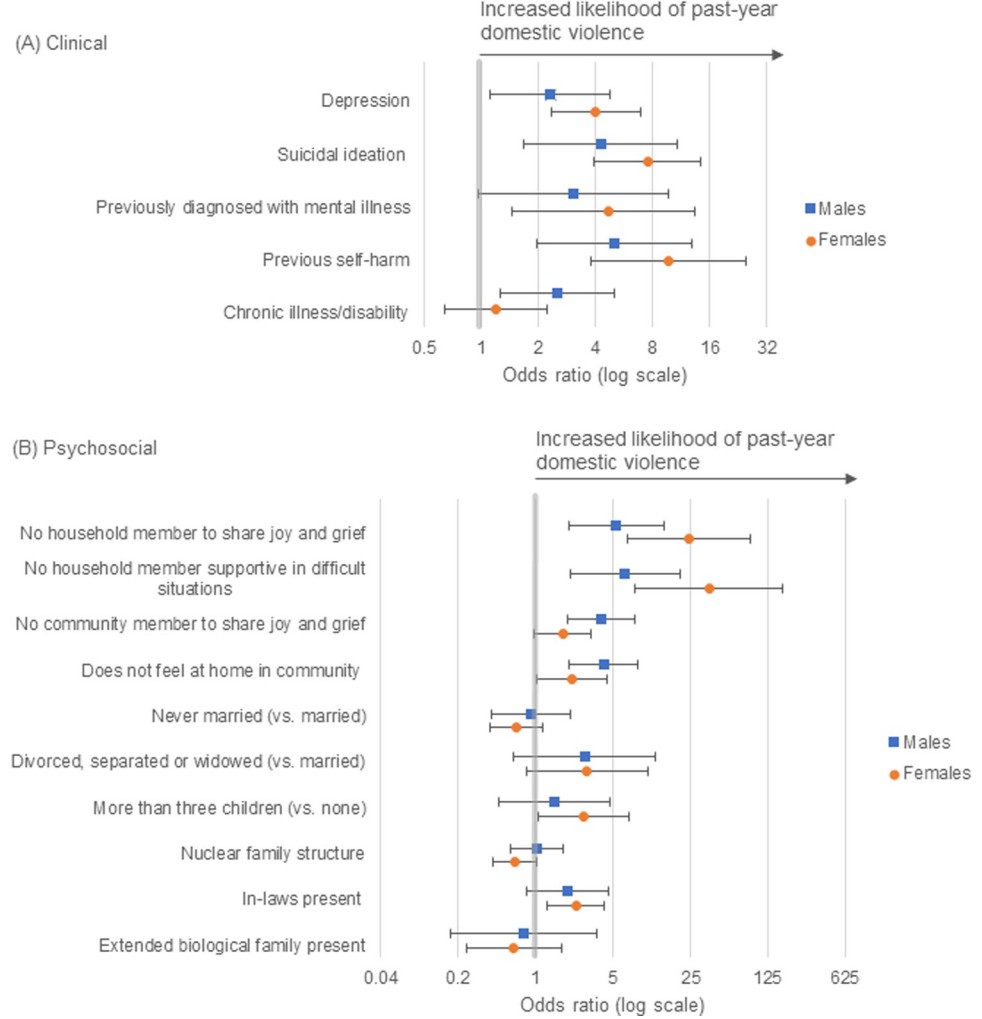

**Fig 2. Clinical and psychosocial factors associated with past-year domestic violence by sex, Kandy, Sri Lanka.** (A) Clinical correlates of past-year domestic violence, adjusted for age. (B) Psychosocial correlates of past-year domestic violence, adjusted for age and ethnicity. The bold line indicates a null result.

The psychological correlates of DV found in the present study are largely consistent with local and international literature [4, 5, 8, 33]. A similar strength in association between depression symptoms and DV has been reported in Bangladesh [34] and India [35]. DV has also been previously shown to be strongly associated with suicidal ideation and self-harm in Asia [17, 36–38]. Although alcohol misuse has been linked with an increased risk of IPV victimisation among men and women in largely high-income countries [39, 40], no association was found in the current study. Alcohol consumption is not socially sanctioned in this context, especially for women [41], which may explain the low numbers reported and absence of an association. This study did not examine alcohol consumption among perpetrators of DV, however, previous studies from Sri Lanka and India have shown alcohol misuse plays an integral part in the perpetration of violence and thus should be an important consideration in DV prevention [42, 43].

Higher frequencies of chronic illness/disability have been reported among women experiencing IPV in other LMIC settings [8, 44, 45]. Contrary to international evidence,

according to Sri Lanka's 2019 Women's Wellbeing Survey, lower rates of past-year IPV have been reported among women with a disability versus no disability [4]. There was no statistical evidence in the current study that chronic illness/disability was associated with DV among women. Given the mixed evidence, further studies are needed to explore how vulnerable populations such as those living with a disability may be affected. Men in the current study with a chronic illness/disability were more likely to report experiencing DV than men without a chronic illness/disability, consistent with reports from the US [46, 47].

Perceived low social support within the household was strongly associated with DV for men and women, and low community social support was also associated with DV among men and to a lesser extent among women. The interplay of DV, poor mental health and social support have been documented in the literature, with social support identified as both mediating and modifying DV outcomes and DV-related mental health outcomes [9, 11]. It is possible social and emotional support may buffer the adverse impacts of DV on mental health. Notably, social support has been found to attenuate the association between past-year DV and hospital presenting self-poisoning in Sri Lanka [17]. Future qualitative and prospective studies are needed to examine the role of social support in the relationship between mental health and DV in this setting.

Social support is likely influenced by household composition. The present study found women living with in-laws were more likely to report DV and this predominantly related to psychological abuse. This is consistent with studies from China [48], Pakistan [49], and Jordan [50]. In contrast to other South Asian countries, dowry-related violence is not common in Sri Lanka [51]. However, it is possible that in-laws may place additional pressure on women to fulfil gender roles and domestic duties. When these expectations are not met, women may be more susceptible to verbal and psychological abuse by household members [3]. Having three or more children compared to no children, was also associated with DV among women, as has been reported in other LMIC [52, 53]. A lack of contraceptive use (and likely women's control over contraceptive use) in conjunction with forced sex has been shown to be associated with DV and subsequently a high number of unwanted pregnancies in India [54–56]. In addition, the presence of more children, may reflect lower socioeconomic position and thus household stress. A previous study in Sri Lanka showed indicators of low socioeconomic position, including lower educational attainment and poor household wealth, increased likelihood of IPV and is likely to interact with clinical and psychosocial factors in this setting and thus an important consideration for DV prevention [57].

## Strengths and limitations

A major strength of this study was the broad criteria for inclusion. There is limited research on the experience of DV among men, particularly in South Asia. This study highlights the importance of recognising DV among men and its clinical implications. In addition, much of the research surrounding DV in South Asia is limited to partnered women of reproductive age. No upper age limit was specified for the present study, and the sample included unmarried, married and previously married or partnered adults. Furthermore, the study included abuse by any family member of the household not just an intimate partner.

Despite this, there are a number of methodological limitations that should be considered when interpreting the data. Firstly, the cross-sectional nature of the study design does not allow for causality to be established. Second, due to logistical and resource constraints, clinical assessments of mental illness and chronic illness/disability were not conducted. An additional limitation to consider is the likely under-reporting of DV, particularly of sexual violence, largely due to socio-cultural factors and potential social desirability bias. In addition, the

HARK questionnaire used to identify DV in the present study has not been validated for use among men and within the Sri Lankan population. Survey instruments have traditionally not been designed to identify abuse among men and thus may not accurately capture male victimisation. In addition, in the context of patriarchal societies (as in Sri Lanka), men may struggle to articulate experiences of abuse, especially if the perpetrator is female. To overcome some of these limitations, the HARK questionnaire was pre-tested with the local population, and comparisons with previous studies show broad consistency in prevalence of abuse for men and women [58].

Furthermore, due to low numbers in some categories (e.g. previously partnered individuals) and overall limitations in the sample size, statistical power was reduced, particularly for stratified analyses and interaction tests. It is possible the lack of statistical differences between men and women may be attributed to the small sample size, disguising actual population level differences in associations by sex. Given participants were recruited based on the age and sex distribution of self-poisoning cases, the sample is predominantly a younger adult population (18–30 years), with women on average younger than men. It is possible that the prevalence of DV in older women and younger men was under-enumerated, distorting the overall prevalence for males and females. However, as previously discussed, estimates were similar to previous studies.

## Implications

It is important to acknowledge that many men and women will not present to hospital for DV or seek specialist services, nor actively seek support for mental health issues. The 2016 Sri Lanka DHS reports less than a third of women (28%) in Sri Lanka will seek help for IPV, and among those that did seek assistance, less than 9% sought help from a health professional. Similarly, according to the 2019 Women's Wellbeing Survey, a fifth of women (21%) who were sexually abused by their partner did not communicate this to anyone [4]. The most common source from which help was sought for IPV was from a family member or friend/neighbour [4, 59]. Given family members, friends and neighbours are important sources of help, and poor social support showed strong associations with DV, raising awareness within community of the importance of social support and of the harmful consequences of DV on mental and physical health may be beneficial. Community-based programs addressing harmful gender norms in Sri Lanka have shown promise [60], and findings from the SASA trial in Uganda showed that a focus on social networks and tackling harmful gender norms was helpful in mobilising community support for survivors of abuse and reducing IPV [61, 62]. The adoption of similar community-based programs may have value in Sri Lanka.

The findings of this study indicate that both men and women experience a similar rate of any DV in Kandy, Sri Lanka. At present, much of the focus has been on the understanding and support of women experiencing DV, but our findings indicate that there is an urgent need to shine a similar focus on men experiencing DV as well. Advancing new and existing awareness campaigns to raise the profile of DV service providers may also be beneficial for clinical populations as well as the wider public. In Sri Lanka, selected hospitals have established gender-based violence support units (*Mithuru Piyasa)* within the outpatient department. However, many of these services and existing campaigns are tailored towards women. In addition, further prospective and qualitative research is needed to understand the context of power dynamics within relationships and how socio-cultural factors differentially affect men and women. Ultimately, a multi-level approach is needed to address individual clinical and psychosocial factors, household factors such as poverty and alcohol misuse, and broader societal factors including gender inequality to address and reduce DV in Sri Lanka.

## Supporting information

**S1 Table. Humiliation, Afraid, Rape, Kick (HARK) questionnaire (English translated version).**
(DOCX)

**S2 Table. Social support questions derived from a social capital community survey in the North Central Province of Sri Lanka.**
(DOCX)

**S3 Table. Sociodemographic characteristics by any exposure to past-year domestic violence (DV).**
(DOCX)

**S4 Table. Clinical and psychosocial factors associated with domestic violence (DV) in Kandy, Sri Lanka, stratified by type of abuse.**
(DOCX)

**S5 Table. Clinical and psychosocial factors associated with domestic violence (DV)–sensitivity analysis restricted to household-based participants in Kandy, Sri Lanka (N = 475).**
(DOCX)

## Acknowledgments

The authors would like to thank the senior academics who have acted as advisors for the study: Professors Chris Metcalfe, and Gene Feder (University of Bristol), Professor Michael Eddleston (University of Edinburgh) and Professor Flemming Konradsen (University of Copenhagen). The authors would like to thank the staff at SACTRC, in particular Chamil Kumara, Indunil Abeyratne, and Sujani Ekanayake for their support in setting up the study and would like to acknowledge the substantial contribution of Azra Aroos, Kasuni Silva, and Sandareka Samarakoon in collecting the data. The authors would also like give thanks to the staff at the Teaching Hospital Peradeniya for accommodating this research, and Mr Upali Perera for designing and maintaining the study database. Acknowledgments also to Dr José López-López and Dr Judi Kidger for providing input into the funding acquisition. The authors would like to acknowledge DG is supported by the NIHR Biomedical Research Centre at University Hospitals Bristol and Weston NHS Foundation Trust and the University of Bristol.

## Author Contributions

**Conceptualization:** Piumee Bandara, Andrew Page, David Gunnell, Duleeka Knipe, Thilini Rajapakse.

**Data curation:** Piumee Bandara, Duleeka Knipe.

**Formal analysis:** Piumee Bandara.

**Funding acquisition:** Lalith Senarathna, Duleeka Knipe, Thilini Rajapakse.

**Investigation:** Piumee Bandara, Tharuka Silva.

**Methodology:** Piumee Bandara, Andrew Page, Lalith Senarathna, Duleeka Knipe, Thilini Rajapakse.

**Project administration:** Piumee Bandara, Duleeka Knipe, Thilini Rajapakse.

**Supervision:** Duleeka Knipe, Thilini Rajapakse.

**Validation:** Duleeka Knipe.

**Visualization:** Piumee Bandara.

**Writing – original draft:** Piumee Bandara.

**Writing – review & editing:** Piumee Bandara, Andrew Page, Lalith Senarathna, Kumudu Wijewardene, Tharuka Silva, David Gunnell, Duleeka Knipe, Thilini Rajapakse.

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
