## [Decision Letter · Decision Letter 0]

2 Nov 2021

PGPH-D-21-00266

Clinical and psychosocial factors associated with domestic violence among men and women in Kandy, Sri Lanka

Dear Dr. Bandara,

Thank you for submitting your manuscript to PLOS Global Public Health. After careful consideration, we feel that it has merit but does not fully meet PLOS Global Public Health’s publication criteria as it currently stands. Therefore, we invite you to submit a revised version of the manuscript that addresses the points raised during the review process.

We look forward to receiving your revised manuscript.

Kind regards,

Rubeena Zakar, Ph.D

Academic Editor

Journal Requirements:

1. Along with your ethics statement, please include a statement that formal consent was obtained (must state whether verbal/written) OR the reason consent was not obtained (e.g., anonymity).

3. We noticed that you used “data not shown”/"unpublished data" in the manuscript. We do not allow these references, as the PLOS data access policy requires that all data be either published with the manuscript or made available in a publicly accessible database. Please either remove these references, or amend the supplementary material to include the referenced data.

4. Please update the completed 'Competing Interests' statement, including any COIs declared by your co-authors. If you have no competing interests to declare, please state "The authors have declared that no competing interests exist". Otherwise please declare all competing interests beginning with the statement "I have read the journal's policy and the authors of this manuscript have the following competing interests:"

5. Since your data is not available for proprietary reasons, please explain via email why the data is not available. Please also include the contact information for the third party organization that should be contacted should other researchers want to request access to this data and please include the full citation of where the data can be found. We also request that you verify with us via email that any researcher will be able to obtain the data set in the same manner that the you have obtained it. If you feel you are unwilling or unable to adhere to this policy, please explain your reasons by return email and your exemption request will be escalated to the editor for approval. Your exemption request will be handled independently and will not hold up the peer review process, but will need to be resolved should your manuscript be accepted for publication. One of the Editorial team will be in touch if they require more information.

Reviewers' comments:

Reviewer's Responses to Questions

**Comments to the Author**

1. Does this manuscript meet PLOS Global Public Health’s publication criteria? Is the manuscript technically sound, and do the data support the conclusions? The manuscript must describe methodologically and ethically rigorous research with conclusions that are appropriately drawn based on the data presented.

Reviewer #1: Yes

Reviewer #2: Partly

2. Has the statistical analysis been performed appropriately and rigorously?

Reviewer #1: Yes

Reviewer #2: Yes

3. Have the authors made all data underlying the findings in their manuscript fully available (please refer to the Data Availability Statement at the start of the manuscript PDF file)?

Reviewer #1: No

Reviewer #2: Yes

4. Is the manuscript presented in an intelligible fashion and written in standard English?

Reviewer #1: Yes

Reviewer #2: Yes

5. Review Comments to the Author

Reviewer #1: The paper analyses the drivers of Domestic violence for men and women in Sri Lanka. The research question is interesting and extremely relevant for public health discourse globally. However, the authors may consider a few changes before publishing this manuscript--

A) the third objective i.e. to explore how the associations may differ by sex and by type of abuse will be better achieved if the model is run on a combined sample of men and women in used and sex is interacted with the key drivers of DV.

B) The model may be repeated for types of abuse to observe if the associations are different.

C) A discussion on the associations of DV and other control indicators (age, education) may be added as there are limited resource on DV on men.

D) Visualization techniques should be explored for narrating the results.

E) There are minor grammatical errors (missing comma etc.).

Overall it is a carefully designed and clearly written article and will make a good addition to the GPH.

Reviewer #2: Summary: the authors examined the association of clinical and psychosocial factors with domestic violence using a dataset from Kandy, Sri Lanka. I commend the authors for looking at this important issue in Sri Lanka and particularly examining the sex difference. However, I have concerns regarding their interpretation of the results and the conclusions they made based on their findings.

Major comments:

• Intro

o In general, the logic flow of the intro needs to be improved. Reading paragraph #2 makes me think this manuscript would mainly focus on men who experienced DV or at least sex difference. However, the manuscript was not written in this way.

o I don’t think the authors could provide prevalence of DV based on the data they used in this paper. It is a convenient sample and the sample size is very small. I would not call the calculated percentage of DV from this sample as prevalence of DV in Sri Lanka.

• Study variables

o When did DV happen? The HARK study limits the timeframe of the violence variables as the past year. Should we think that this manuscript used the same method? This section needs to be revised to add more clarity and transparency.

• Discussion

o Again, I don’t think this current study could provide prevalence of DV in Sri Lanka. The language needs to be revised.

o “A key finding was that men and women experienced DV to a similar degree in Sri Lank”. This is a very vague conclusion. What does similar mean here? The findings only suggested in the sample there is no significant difference between men and women in terms of the association between psychological factors and DV. This conclusion was made well beyond the findings.

o Page 14, line 31: “notably, the prevalence of DV was similar for men and women.” I didn’t find the language in the results section.

o Table 4, for the social support factors, we actually see big a difference between females and males despite statistical insignificance. Note that, p-values are all at boarder line and the small sample size could contribute to that. It would be inappropriate to conclude that there is no sex difference of the associations in Sri Lanka.

o Page 15, line 43: the interpretation of the alcohol consumption finding is not clear.

o Page 15, line 50: the cited finding does not support the manuscript’s finding. The cited findings is that women with a disability had lower IPV rate than those without disability. However, the authors just found no statistical evidence between disability and DV among women. The cited finding is not the same as authors’ finding.

Minor comments

• Intro

o Paragraph #2: what does “to a similar degree” mean specifically?

6. PLOS authors have the option to publish the peer review history of their article (what does this mean?). If published, this will include your full peer review and any attached files.

**Do you want your identity to be public for this peer review?** For information about this choice, including consent withdrawal, please see our Privacy Policy.

Reviewer #1: No

Reviewer #2: No

---

## [Decision Letter · Decision Letter 1]

13 Feb 2022

Clinical and psychosocial factors associated with domestic violence among men and women in Kandy, Sri Lanka

PGPH-D-21-00266R1

Dear Dr Bandara,

We are pleased to inform you that your manuscript 'Clinical and psychosocial factors associated with domestic violence among men and women in Kandy, Sri Lanka' has been provisionally accepted for publication in PLOS Global Public Health.

Best regards,

Rubeena Zakar, Ph.D

Academic Editor

Reviewer Comments (if any, and for reference):

Reviewer's Responses to Questions

**Comments to the Author**

1. If the authors have adequately addressed your comments raised in a previous round of review and you feel that this manuscript is now acceptable for publication, you may indicate that here to bypass the “Comments to the Author” section, enter your conflict of interest statement in the “Confidential to Editor” section, and submit your "Accept" recommendation.

Reviewer #1: All comments have been addressed

Reviewer #2: All comments have been addressed

Reviewer #3: All comments have been addressed

2. Does this manuscript meet PLOS Global Public Health’s publication criteria? Is the manuscript technically sound, and do the data support the conclusions? The manuscript must describe methodologically and ethically rigorous research with conclusions that are appropriately drawn based on the data presented.

Reviewer #1: Yes

Reviewer #2: Yes

Reviewer #3: Yes

3. Has the statistical analysis been performed appropriately and rigorously?

Reviewer #1: Yes

Reviewer #2: Yes

Reviewer #3: Yes

4. Have the authors made all data underlying the findings in their manuscript fully available (please refer to the Data Availability Statement at the start of the manuscript PDF file)?

Reviewer #1: Yes

Reviewer #2: Yes

Reviewer #3: Yes

5. Is the manuscript presented in an intelligible fashion and written in standard English?

Reviewer #1: Yes

Reviewer #2: Yes

Reviewer #3: Yes

6. Review Comments to the Author

Reviewer #1: (No Response)

Reviewer #2: (No Response)

Reviewer #3: Dear Authors,

I appreciate your work. Well done!

7. PLOS authors have the option to publish the peer review history of their article (what does this mean?). If published, this will include your full peer review and any attached files.

**Do you want your identity to be public for this peer review?** For information about this choice, including consent withdrawal, please see our Privacy Policy.

Reviewer #1: **Yes: **Ruchira Bhattacharya

Reviewer #2: No

Reviewer #3: No
